# Assessment of the Micro-Tensile Bond Strength of a Novel Bioactive Dental Restorative Material (Surefil One)

**DOI:** 10.3390/polym16111558

**Published:** 2024-05-31

**Authors:** Abdulrahman A. Alghamdi, Smaher Athamh, Reham Alzhrani, Hanan Filemban

**Affiliations:** 1Restorative Dentistry Department, Faculty of Dentistry, King Abdulaziz University, Jeddah 21589, Saudi Arabia; hfilemban@kau.edu.sa; 2Faculty of Dentistry, King Abdulaziz University, Jeddah 21589, Saudi Arabia; sathamh@stu.kau.edu.sa (S.A.); ralzhrani0129@stu.kau.edu.sa (R.A.)

**Keywords:** Surefil one, bioactive, micro-tensile bond strength, self-adhesive

## Abstract

Objectives: The aim of this study is to assess the micro-tensile bond strength and the mode of failure of a bioactive hybrid self-adhesive composite (Surefil one) under various dentin conditions. Methods: Thirty-two extracted human molar teeth were used to test the micro-tensile bond strength of Surefil one under different dentine conditions (no treatment, 37% phosphoric acid etching, and universal adhesive) in comparison with a resin-modified glass ionomer (RIVA). All restorations were light cure-bonded onto flat dentine and then sectioned into beams. Then, fractured specimens were observed under a light microscope to evaluate the mode of failure. Results: The Surefil one no-treatment group (NTG) exhibited the highest micro-tensile bond strength. Furthermore, there was no statistically significant difference observed between the Surefil one adhesive group (EAG) and the Surefil one acid etch group (EG). However, compared to other groups, the resin-modified glass ionomer (RIVA) produced the lowest results, which are statistically significant. Conclusion: Surefil one offers superior bond strength values when compared to resin-modified glass ionomers. Furthermore, Surefil one requires no dentin condition and has more straightforward clinical steps.

## 1. Introduction

Resin-based composite restorations were developed in 1964 and were one of the biggest innovations in esthetic dentistry. Several research studies indicated that the dental resin-based composite has excellent mechanical properties, including superior hardness, high wear resistance, less water solubility, and biological compatibility [1]. Furthermore, such a material necessitates knowledge and experience since it is technique-sensitive and requires a proper surface treatment in advance, including proper isolation, cavity design, adhesive application, and incremental layering, to ensure proper polymerization for better clinical results. In order to simplify the clinical procedure in terms of time and technical sensitivity, researchers have introduced self-adhesive bulk-fill restorative materials such as glass ionomer cement (GIC), resin-modified glass ionomer (RMGI), and self-adhesive composite. Glass ionomers have several applications, for example, as a restorative material, sealant, luting cement, and cavity liner [2]. However, with time, GIC usage in clinical settings was reduced as a consequence of low esthetic appearance, along with its low flexural strength and low wear resistance [3]. On the other hand, RMGIs are produced by adding resin fillers to the glass ionomer, aiming for enhancement of the mechanical properties, less setting time, and moisture sensitivity. Nevertheless, GIC and RMGIs still have a low initial strength and sensitivity to moisture during the setting reaction [4].

Several studies were developed to overcome the unpleasant esthetic appearance and the low mechanical properties of the two previously mentioned materials. For instance, a recent development of a new self-adhesive composite known as Surefil one (Dentsply Sirona, Konstanz, Germany) seems promising. This material has superior mechanical properties over GIC and RMGIs and comparable mechanical behaviors to amalgam [5]. Additionally, Surefil one aims to reduce the technical sensitivity and time-consuming steps associated with resin-based composites [6]. Surefil one has similar chemical components that are present in conventional resin-based composites, including structural monomers, reactive diluent, silanated glass particles, and sharing fluoride release and the self-adhesive properties of polyacids that are present within GIC. The distinctive chemical feature in Surefil one that contributed to its self-adhesive property is the presence of the functional modified polyacid system (MOPOS) [7]. Surefil one bonds chemically and micromechanically to the tooth structure by utilizing the carboxylic acid groups in both MOPOS and acrylic acid to bind calcium ions of the hydroxyapatite and through the infiltration of the smear layer and surface demineralization or hybridization [8]. As a result, Surefil one would be a great material of choice for situations where time is limited or when dealing with young patients who may not be able to cooperate for an extended period of time [9].

As with any new material, clinical and in vitro studies to test the biological, mechanical, durability, and optical properties were evaluated. Multiple clinical studies found Surefil one to be promising [7,10,11]. According to Lohbauer and Belli’s evaluation, Surefil one’s mechanical characteristics fell within a range that was comparable to that of the resin-based composites when compared to three direct composite resins and two GIC materials. Another essential point to assess the efficiency of any restorative material is by measuring either or both tensile and shear bond strength [12]. To date, there are a few reports in the literature about the micro-tensile bonding strength of Surefil one. In addition, most of the reports measure the micro-tensile bonding strength material under one dentin condition.

So, the aim of this in vitro study is to evaluate the micro-tensile bond strength of the Surefil one self-adhesive bioactive material and examine whether different dentin conditions can have an impact on its adhesion.

The null hypotheses were:There are no differences in the micro-tensile bond strength between Surefil one and RIVA.Different dentin conditions have no effect on Surefil one’s micro-tensile bond strength.

## 2. Materials and Methods

This in vitro study was performed using protocols approved by the Ethics Committee of King Abdulaziz University in Saudi Arabia, Jeddah, approval number 4469514. This in vitro study included a new self-adhesive restorative material, Surefil one, and resin-modified glass ionomer, RIVA. The composition and application technique of the investigated materials and all chemicals used are presented in Table 1 and Table 2, respectively.

### 2.1. Specimens and Sampling Technique

Thirty-two human-extracted sound molar teeth were collected and cleaned by using a scaler to remove all debris and staining and then stored in 10% formalin. The samples were selected by specific inclusion and exclusion criteria, which are: Inclusion criteria: Teeth with sound and caries-free dentine; molar teeth.Exclusion criteria: Endodontically treated teeth; badly decayed teeth; premolar and anterior teeth.

The samples were randomly split into four groups (8 teeth per group) using a simple random sampling technique as follows: Surefil one no-treatment group (NTG): Surefil one was applied directly to the dentine surface and light-cured for 20 s.Surefil one acid etch group (EG): Dentin was acid etched using 37% phosphoric acid for 15 s before the application of Surefil one and light-cured for 20 s.Surefil one adhesive group (EAG): Dentin was acid etched using 37% phosphoric acid for 15 s; then, universal adhesive was applied using a micro brush and spread gently with air and light-cured for 20 s, followed by Surefil one application.Resin-modified glass ionomer control group (RIVA): The material was applied directly to the dentine surface and light-cured for 20 s.

### 2.2. Specimen Preparation

For the micro-tensile test, thirty extracted molars were used. Teeth were mounted by the roots in cold-cured acrylic to help in handling the samples, then cut horizontally by a low-speed diamond saw sectioning machine (ALLIED, Cerritos, CA, USA) with a speed of 380 RPM under constant water cooling to expose the flat dentin surface. The dentin surface was wet-sanded with 600 grit sandpaper for 1 min to create a smear layer that simulates diamond bur cuts. Specimens were divided into four groups, as mentioned above. Surefil one has been tested by directly applying the material to the dentin surface without any previous dentin treatment (NTG), then compared with two other conditions, which are 37% phosphoric acid (EG) and universal adhesive with total etching (EAG), and resin-modified glass ionomer (RIVA) as the control group. In all groups, Surefil one and RIVA were applied to the dentine surface, adapted using a plastic filing instrument, and then light-cured for 20 s according to the manufacturer’s instructions using a 3M (Paradigm DeepCure LED, Saint paul, MN, USA) curing light. All specimens were placed in normal saline and stored in an incubator at 37 °C for 24 h. The steps are presented in Figure 1.

### 2.3. Micro-Tensile Strength Measurements and Failure Modes

After 24 h, each tooth was sectioned vertically in two directions into beams (0.8 mm × 0.9 mm, ±0.3); each beam was measured using a digital caliber. The beams were mounted in a micro-tensile jig using a cyanoacrylate adhesive (Zapit, DVA, Corona, CA, USA). Each jig was placed in a universal testing machine (Shimadzu EZ-SX, Kyoto, Japan) at a speed of 1 mm/minute, Failure strength (MPa) was calculated using the following formula:$$\frac{\left[N\right]}{\left[area\left({mm}^{2}\right)\right]}$$
where N is the failure load in newton, and area is the cross-section of the sample in mm^2^.

All fractured specimens were evaluated visually to observe the mode of failure; then, selected specimens were chosen according to their subgroups, which included the highest and lowest values in addition to in-between values and pre-test failures. These subgroups were analyzed under a stereomicroscope (Plan 10/0.25; 160/0.17) to detect the mode of failure, and both sides of the specimen (dentin side and restoration side) were evaluated to determine the overall mode of failure. The following were the failure categories:Adhesive: Interfacial bond failure between the restoration and dentine.Cohesive in the restoration: When a fracture allows a layer of restoration to remain on both surfaces.Cohesive in the dentin: When a fracture allows a layer of dentine to remain on both surfaces.Mixed failure: If the failure shows more than one substrate [13].

A visual illustration of the different modes of failure is presented in Figure 2.

### 2.4. Statistical Analysis

Statistical analyses were performed using JMP Pro 15 statistical software (SAS Institute, Cary, NC, USA). Normal distribution was confirmed by the Shapiro–Wilk test, and equality of variance was assessed using Levene’s test before the tests were performed. The analysis of differences between the study groups was performed using a one-way analysis of variance (ANOVA). Once significant differences were detected, post-hoc Tukey–Kramer multiple comparison tests were used to compare means. Statistical significance was set at *p* < 0.05.

## 3. Results

Micro-tensile bond strength (μTBS) was measured and summarized for all specimens (Table 3). Initially, all groups of Surefil one were compared independently without RIVA to evaluate the effect of different dentin conditions (Figure 3). Afterward, each group of Surefil one was compared to RIVA to assess the variations in the mean values of μTBS.

In general, the Surefil one groups had the greatest μTBS values, whereas the RIVA control group had the lowest values. This variation was statistically significant (*p* < 0.05). In contrast, the μTBS values for the Surefil one groups ranged from 18.42 to 16.07 MPa, depending on the dentin condition. The highest value was measured for the NTG with 18.42 ± 7.96 MPa, followed by the EG with 16.13 ± 7.33 MPa, and the EAG was 16.07 ± 6.27 MPa. However, according to the one-way ANOVA test, there was no significant difference among these groups (*p* < 0.0001). However, when compared to the RIVA group, it was significantly different for all Surefil one groups using the Tukey test presented in Table 4.

Regarding the mode of failure results, Figure 4 shows a representative sample reading. The NTG showed 40% adhesive failure, 25% cohesive failure in dentine, 30% cohesive failure in restoration, and 5% mixed failure. In the EG, the failure percentages were 21.87%, 37.5%, 18.76%, and 21.87, respectively. The EBG showed failure percentages of 25.53%, 14.89%, 53.19%, and 6.38%, respectively. In the RIVA control group, the failure percentages were 77.7%, 5.6%, 13.9%, and 2.8%, respectively, as shown in Figure 5.

## 4. Discussion

Currently, resin-based composite restorations are the most popular restorative materials utilized in the world, mostly due to their excellent mechanical and aesthetic qualities. Nevertheless, the yearly failure rate of composite restoration is 2.4% after ten years and 1.8% after five. The two primary reasons for failure are fractures and recurrent caries. As previously known, several factors play important roles in recurrent caries, such as the clinical expertise of the operators, moisture control, polymerization shrinkage, oral hygiene, and diet, in addition to the cooperation of the patient during the procedure [14]. In order to reduce the technical sensitivity of composite restorations and simplify the clinical procedure, GIs, RMGIs, and self-adhesive resin-based composites are still desirable. Recently, a hybrid self-adhesive composite was introduced on the market—Surefil one (Dentsply-Sirona, Charlotte, NC, USA). This material combines the good characteristics of the most commonly used restorative material: composite resin esthetic and mechanical properties and conventional glass ionomer self-adhesion and fluoride release. Surefil one showed promising mechanical properties, fluoride release, and equal or better shear bond strength than GIs and RMGIs [15]. However, not many studies on its micro-tensile bond strength have been found.

Surefil one seems promising due to its impressive self-adhesive properties, which are a direct result of its unique chemical composition, specifically, the inclusion of a modified polyacid MOPOS within the formula that is responsible for both its strong self-adhesion and mechanical durability. MOPOS has the ability to dissolve the smear layer and underlying mineral components of both enamel and dentin while also facilitating chemical bonding through its acidic monomers. Furthermore, MOPOS acts as a copolymerizing cross-linker between covalent and ionic networks, making it an incredibly effective and reliable bonding agent [7]. To ensure the effectiveness of Surefil one, it is important to test its bond strength on dentin surfaces. In this study, we have chosen the micro-tensile test to evaluate the bond strength of Surefil one to dentin. This test has been selected because of its consistent and reliable results. The experimental study was performed by dividing thirty teeth into four groups. Three of them were assigned to test the micro-tensile bond strength of Surefil one under different dentin conditions, and the last group was used as control, which was restored with RIVA (RMGI). 

In addition, one of the purposes of this in vitro study is to measure if the micro-tensile bond strength of Surefil one will be improved after different dentine treatments, including 37% phosphoric acid etch with and without universal bonding.

In the micro-tensile bond strength test, the RIVA control group had lower values in the micro-tensile bond strength test compared to all of the Surefil one groups. Thus, the first hypothesis was rejected as the μTBS results were significantly different between the two groups. In addition, the RIVA group had a considerable number of pre-test failures. This reduction in micro-tensile bond strength can be attributed to the material’s high solubility and water sorption, which could potentially cause a decrease in the bond strength of the material [16,17].

Additionally, this study examined whether different dentin conditions would affect the Surefil one micro-tensile bond strength values. The results showed that when comparing the μTBS values between the Surefil one groups, the NTG showed higher μTBS results compared to the EG and the EAG. However, the difference was not significant, and thus the second hypothesis was accepted. Although the difference was not significant, we can speculate that this improvement in micro-tensile values can be attributed to the ability of Surefil one to integrate all primary adhesion mechanisms, such as surface wetting, chemical covalent bonding, and micromechanical interlocking, resulting in improved adhesion performance even without any pre-dentin treatment [18]. For instance, the aqueous nature of Surefil one allows for the surface to be sufficiently wet without introducing excessive moisture, thereby avoiding the transformation of the tooth’s surface from hydrophilic to hydrophobic, which is an essential part of self-adhesive bonding. In addition, the acidic monomers (MOPOS) contained within Surefil one (Figure 6) create a primary ionic bond with the hydroxyapatite of the tooth substrate by modifying the smear layer surface of both dentin and enamel, thereby facilitating micro-retention [9]. Furthermore, the distinct bifunctional acrylate (BADEP) monomer’s reactive double bond groups establish a direct link between the ionic and covalent networks, producing a cohesive, extremely stable molecular network that strengthens bonds [19]. These findings give the overall impression that this new material is best used without any modification to the dentin surface since etching dentin and adhesive application did not add any value to the micro-tensile bond strength. On the contrary, it seems like the etchant and adhesive steps slightly reduce the bonding strength values. There has only been a few studies on the μTBS of Surefil one, as evidenced by references [20,21,22]. Our own μTBS findings align with those of Yao C. et al. and A. Elraggal et al. However, the latter study observed an increase in μTBS with the addition of a universal adhesive, which could be attributed to their use of a self-adhesive technique rather than the total-etch method employed in our own research. In a recent study conducted by Eichler E. et al., the bond strength of Surefil one was evaluated after 24 h of water storage. Surefil one had a value of 2.7 ± 0.5 MPa after 24 h of water storage, which is considered poor in terms of bond strength. This could be attributed to the fact that the dentin surface was kept moist during the conditioning process, which is unnecessary since the Surefil one already contains enough water to achieve proper surface wetting and adhesion. 

Based on the failure results, it has been observed that the NTG has a higher adhesive failure rate than the other modes. Additionally, in the case of the EG, the etching step does not seem to provide benefits as it results in more dentin cohesive failure. This could be attributed to the phosphoric acid’s potential to eliminate minerals and leave collagen fibrils without support. Additionally, no hybrid layer was created to provide support and retain the restoration. Nevertheless, based on the findings, it has been observed that the hybrid layer formed by the EAG effectively supported collagen fibrils and reduced cohesive failure in dentin. However, it was found that neither the EG nor the EAG significantly reduced adhesive failure, which is in line with the results of the micro-tensile test. Consequently, we can conclude that the etchant and adhesive steps do not offer any noticeable advantages or disadvantages to the bond strength of Surefil one, and their inclusion or omission does not appear to have a significant effect on the results. Furthermore, pre-test failures were identified in the control RIVA group, which can be attributed to weak bonding between dentine and the restoration and led to the majority of adhesive failures in this group.

The resin-modified glass ionomer has been selected as the control material for our experiment due to its remarkable ability to establish a chemical bond with both dentin and enamel surfaces. Despite its minor shortcomings, such as challenges in penetrating the dentin surface and reduced bond strength, this material has gained popularity due to its exceptional bioactivity and capacity to release and recharge fluoride ions [16,17,23].

The study we conducted has demonstrated that Surefil one possesses an impressive capacity to bond with dentin, exhibiting high micro-tensile bond strength values. Furthermore, the potential of Surefil one to release fluoride is highly promising. However, it is important to note that this study was limited in its scope as it was conducted “in vitro” and therefore did not allow for the presence of various elements present in the oral cavity, such as saliva, salivary proteins, different thermal temperatures, and bacteria, to affect the restorative process. As such, there is a need for further investigations to evaluate Surefil one’s long-term bonding durability to the dentin surface under more realistic conditions, including thermal cycling. Therefore, it is imperative to consider the inherent limitations of this “in vitro” study when interpreting the results.

## 5. Conclusions

Within the limitations of this study, the following can be concluded:

-Surefil one has a higher micro-tensile bond strength than RIVA.-Different dentin conditions did not significantly affect Surefil one’s μTBS performance.-Surefil one could potentially be used as a substitute for RMGI restorative material, especially in cases of long-term temporary restorations, since it has a better tensile bond and fluoride release, and there is no need for added complicated steps.-The restoration is not technique-sensitive and is less time-consuming.-Further long-term clinical and laboratory studies are needed.

## Figures and Tables

**Figure 1 polymers-16-01558-f001:**
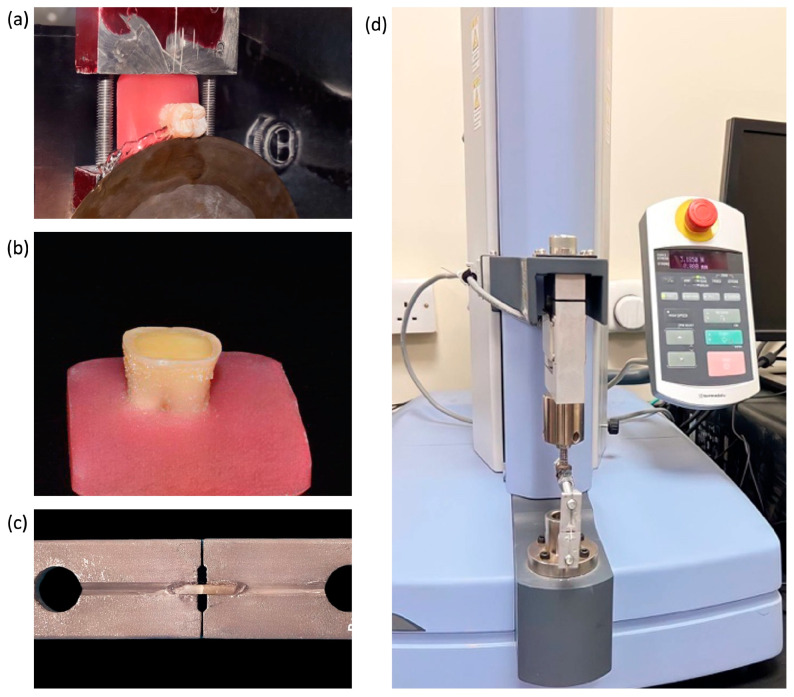
A schematic illustration of micro-tensile specimen preparation. (**a**) Sectioning the teeth to expose flat dentin; (**b**) flat dentin is sanded to receive the restoration; (**c**) sectioned beams mounted on the micro-tensile jig; (**d**) jig is mounted on the universal testing machine to test it.

**Figure 2 polymers-16-01558-f002:**
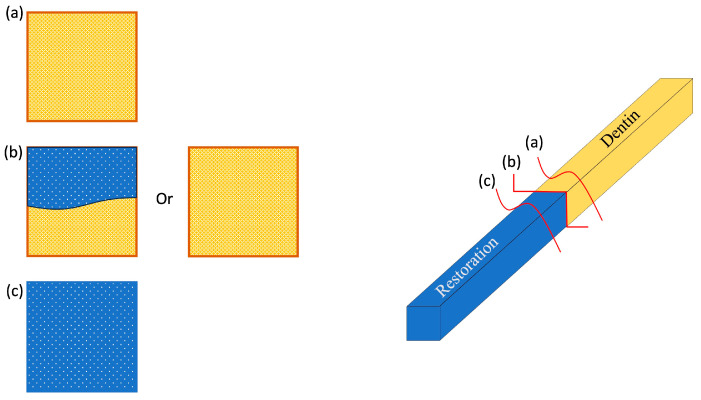
Illustration of the mode of failure. (**a**) Cohesive failure in dentin; (**b**) The adhesive interface that has two different scenarios, which are mixed failure and adhesive failure; (**c**) A cohesive failure in restoration.

**Figure 3 polymers-16-01558-f003:**
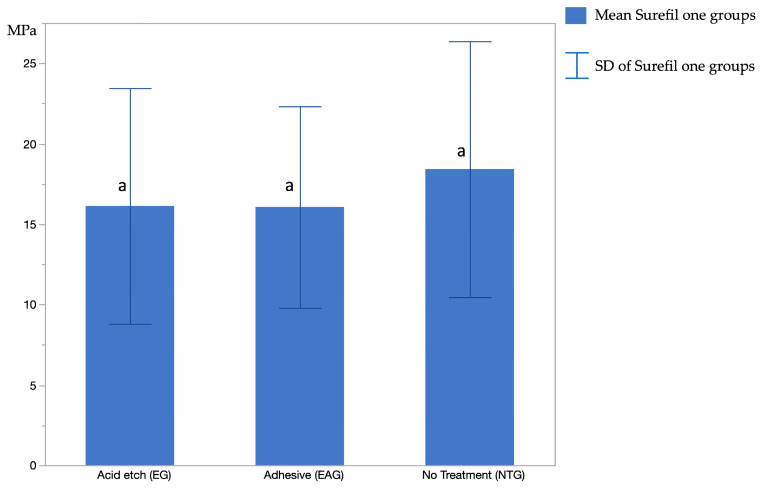
The mean micro-tensile bond strength (MPa) of Surefil one groups. Same-level letters indicate insignificance.

**Figure 4 polymers-16-01558-f004:**
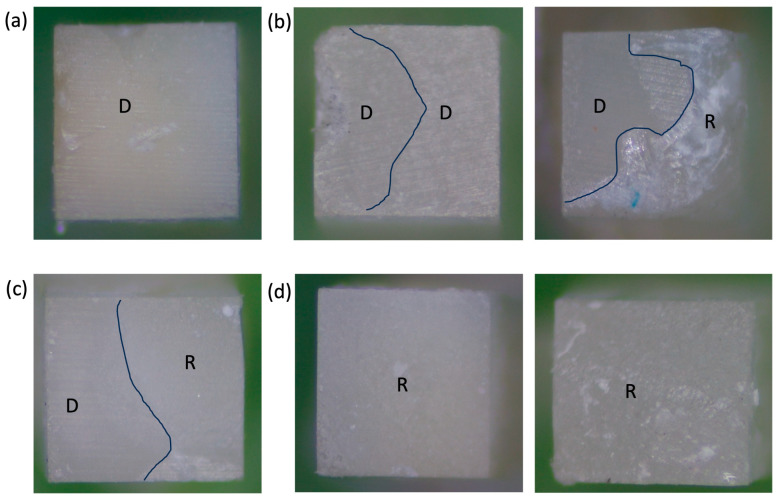
Representative of failure mode samples. (**a**) Adhesive failure; (**b**) Cohesive failure in dentin on the left and the shipped part of dentin on the right sample; (**c**) Mixed failure; and (**d**) Cohesive failure in restoration on both sides of the samples. R = restorative materials; D = dentin.

**Figure 5 polymers-16-01558-f005:**
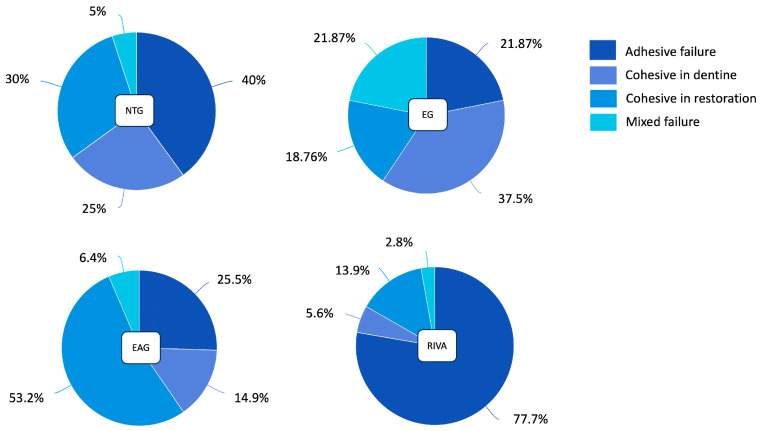
Mode of failure of all groups.

**Figure 6 polymers-16-01558-f006:**
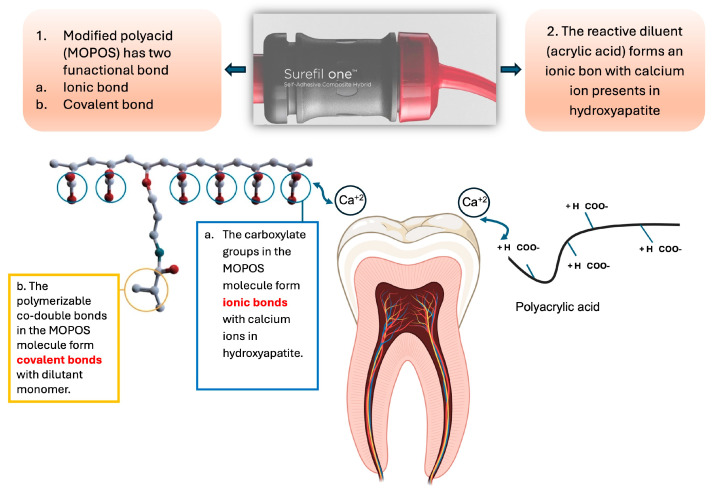
A schematic diagram illustrating the chemical interaction between the Surefil one chemical component and the tooth substrate.

**Table 1 polymers-16-01558-t001:** Restorative materials and adhesives used in this study.

Material, Manufacturer	Composition	Type	Application Technique
Surefil one, Dentsply Sirona, Charlotte, NC, USA	Modified polyacid (MOPOS), bifunctional acrylate (BADEP), acrylic acid, reactive glass filler, water, non-reactive glass filler, initiator, and stabilizer	Self-adhesive bulk Fill	Activate the capsule by pressing it onto a stable surface; then, mix it in the amalgamator for 10 s; then, place it by using a capsule dispenser in the deepest part of the cavity; then, light cure for 20 s
RIVA resin-modified glass ionomer, SDI, Victoria, Australia	Ion glass filler, fluoride, strontium ions photo-initiators, polyacrylic acid, water, and water-soluble methacrylate monomer	Self-adhesive RMGI	Activate the capsule by pressing it onto a stable surface; then, mix it in the amalgamator for 10 s; then, place it by using a capsule dispenser in the deepest part of the cavity; then, light cure for 20 s (use the layering technique for cavities deeper than 2 mm)
Prime&Bond active, Dentsply Sirona, Charlotte, NC, USA	Bi- and multi-functional acrylate, modified phosphoric acid, acrylate resin, initiator, stabilizer, and isopropanol	Universal adhesive	Apply a bonding agent to all surfaces and slightly agitate for 20 s; then, evaporate the solvent using air for at least 5 s; then, light cure for 10–20 s depending on the power output: ≥500 mW/cm^2^ =20 s ≥800 mW/cm^2^ = 10 s

**Table 2 polymers-16-01558-t002:** Chemicals used in this study.

Material, Manufacturer	Composition	Instruction	Usage
Formaline 10%, Thermo Fisher Scientific Inc., Waltham, MA, USA.	Paraformaldehyde, RO water, sodium hydroxide, and HCL	25 g Para-formaldehyde mixed with 250 mL/RO water and 3 full drops/sodium hydroxide. Stir with heat until clear and then buffer to neutral pH with HCl	To sterilize the teeth and as storage media
Zapit glue, Dental Ventures of America Inc., Corona, CA, USA.	Ethyl-2 Cyanoacrylate: Poly (Methyl Methacrylate) Hydroquinone	Place a small amount of glue on each side of the specimen and then spray the accelerator to speed up the setting time	To glue the specimens into the micro-tensile jigs
37% Phosphoric acid, FGM, Joinville, Brazil.	Water-based gel containing 37% phosphoric acid	Etch the enamel and dentin for 15 s; then, wash the surface abundantly with water and dry the cavity	To remove the smear layer

**Table 3 polymers-16-01558-t003:** Mean and SD μTBS of all groups.

			Micro-Tensile, MPa	
Restoration	Dentin Condition	N	Mean	Standard Deviation
Riva control	Control	36	7.83	±7.41
Surefil one	Acid etch (EG)	32	16.13	±7.33
	Adhesive (EAG)	47	16.07	±6.27
	No treatment (NTG)	40	18.42	±7.96

**Table 4 polymers-16-01558-t004:** Tukey HSD all pairwise comparisons.

Restoration	Dentin Condition	Restoration	Dentin Condition	Prob > |t|
RIVA	Control	Surefil one	Adhesive	<0.0001 *
RIVA	Control	Surefil one	No treatment	<0.0001 *
RIVA	Control	Surefil one	Acid etch	<0.0001 *
Surefil one	Adhesive	Surefil one	No treatment	0.4302
Surefil one	Adhesive	Surefil one	Acid etch	1.0000
Surefil one	No treatment	Surefil one	Acid etch	0.5402

* Show pairs of means that are significantly different.

## Data Availability

Data are contained within the article.

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
