# Peer review of "Assessment of the Micro-Tensile Bond Strength of a Novel Bioactive Dental Restorative Material (Surefil One)"

_polymers, 2024, doi:10.3390/polym16111558_

Round 1

Reviewer 1 Report

Comments and Suggestions for Authors

The article is relatively easy to read and the methods part of work is rather well described.

My comments are bulleted below.

4.      Introduction – sentence: “Furthermore, to overcome the technical sensitivity and time-consuming steps in comparison to resin composite” should be rearranged.

5.      The hyperlink to tables and figures does not work.

6.      What was the size of samples? How many samples was for each tested group?

7.      Riva has microtensile bond strength mean higher than standard deviation? (Table 3) this result is unacceptable in my opinion. Sample amount should need to be increased to accept the result of SBS dentine-RIVA.

8.      For this reason, the comparison of the obtained results to μTBS of RIVA-dentine seems to me, at least, uncertain. Would it be possible to compare the obtained results with other similar studies?

9.      Additional figure of proposed chemical mechanism of connection between monomers from Surefill one and minerals/chemicals from dentine would be nice since this is POLYMERS journal.

10.  There was also lack of explanation why μTBS of Surefill one-dentine is higher without any treatment?

11.  What value of μTBS is clinical reliable?

12.  It is customary to test the joints of two materials also after thermocyclic loading. Why didn't the Authors do this? Please also highlight this fact as "research limitations"

Reviewer 2 Report

Comments and Suggestions for Authors

The article entitled 'Assessment of micro-tensile bond strength of novel bioactive restorative material' looks interesting, but needs a few improvements:

1. In my opinion, there is a lack of a clear bullet point in the introduction of the novelty of the paper and a clear aim of the research. 

2. Incorrect identification of citation sources in the text. References should be before the full stop, not after - this is an editing error. 

3. In the paper there is a noticeable lack of comparison to other references in a similar subject area. I recommend checking publications such as: https://doi.org/10.3390/ma15165577 or another 

 4. Edit table 1 and 2 - redundant rows - please delete. 

5. The formula on line 124 should be labelled with symbols, not descriptive, and an explanation of the symbols should be given below. 

6. In line 130, the authors give categories of damage in adhesive joints. There is no reference to the literature here. It would be useful to add example drawings or photographs of the differences between the different types of damage. 

7. The quality of the drawings in the work is poor. Especially figure 1 and 2. They need to be improved and adapted to the editing requirements. 

8. Line 150, 164 - Error! Reference source not found - citation error.

9. Figure 2 - what do the bars and whiskers mean? Please explain in the text or in the legend of the diagram. 

10. The results lack photographs of the damage to the specimens tested. The results in Figure 3 are unreliable without depicting at least an example of the destruction of the samples. 

11. Linguistic correctness needs to be looked at. Much of the phrasing is inappropriate. Example - Glued is not a technical phrase. It is worth checking and correcting the nomenclature.

12. Conclusions would be clearer if they were presented in bulleted form. They should also be more elaborated, as in their current form they resemble the summary of a report of a very short study rather than the conclusions of a scientific paper. 

Round 2

Reviewer 2 Report

Comments and Suggestions for Authors

The article entitled: Assessment of Micro-Tensile Bond Strength of Novel Bioactive Dental Restorative Material (Surefil One) has been revised according to the comments, but not all comments have been taken into account. 

- e.g. the units are still not properly notated, such as in square millimetres 2 is not in subscript, e.g. line 133

- Punctuation errors, e.g. line 61, 145.